# Expression of Prostaglandin Genes and β-Catenin in Whole Blood as Potential Markers of Muscle Degeneration

**DOI:** 10.3390/ijms241612885

**Published:** 2023-08-17

**Authors:** Anna Wajda, Diana Bogucka, Barbara Stypińska, Marcin Jerzy Radkowski, Tomasz Targowski, Ewa Dudek, Tomasz Kmiołek, Ewa Modzelewska, Agnieszka Paradowska-Gorycka

**Affiliations:** 1Department of Molecular Biology, National Institute of Geriatrics, Rheumatology and Rehabilitation, 02-637 Warsaw, Poland; diana.hasan@spartanska.pl (D.B.); barbara.stypinska@spartanska.pl (B.S.); ewa.dudek@spartanska.pl (E.D.); tomasz.kmiolek@spartanska.pl (T.K.); ewa.modzelewska@spartanska.pl (E.M.); agnieszka.paradowska-gorycka@spartanska.pl (A.P.-G.); 2Department of Geriatrics, National Institute of Geriatrics, Rheumatology and Rehabilitation, 02-637 Warsaw, Poland; marrad.md@gmail.com (M.J.R.); tomasz.targowski@spartanska.pl (T.T.)

**Keywords:** sarcopenia, frailty syndrome, prostaglandin, β-catenin

## Abstract

Prostaglandin signaling pathways are closely related to inflammation, but also muscle regeneration and processes associated with frailty and sarcopenia, whereas β-catenin (*CTNNB1* gene) as a part of Wnt signaling is also involved in the differentiation of muscle cells and fibrosis. The present study analyzed the association between selected prostaglandin pathway genes and clinical parameters in patients with sarcopenia and frailty syndrome. The present study was conducted on patients with sarcopenia, frailty syndrome, and control older patients (N = 25). Additionally, two healthy controls at the age of 25–30 years (N = 51) and above 50 years old (N = 42) were included. The expression of the *PTRGER4*, *PTGES2* (*COX2*), *PTGS2*, and *CTNNB1* genes in whole blood was checked by the qPCR method. The serum cytokine levels (IL-10, TNFα, IFN-y, IL-1α, IL-1β) in patients and controls were checked by the Q-Plex Human Cytokine Panel. The results showed a significant effect of age on *PTGER4* gene expression (*p* = 0.01). A negative trend between the appendicular skeletal muscle mass parameter (ASSM) and the expression of *PTGER4* has been noted (r = −0.224, *p* = 0.484). *PTGES2* and *PTGS2* expressions negatively correlated with creatine phosphokinase (r = −0.71, *p* = 0.009; r = −0.58, *p* = 0.047) and positively with the functional mobility test timed up and go scale (TUG) (r = 0.61, *p* = 0.04; r = 0.63, *p* = 0.032). In the older control group, a negative association between iron levels and the expression of *PTGS2 (r =* −0.47, *p* = 0.017) was observed. A similar tendency was noted in patients with sarcopenia (r = −0.112, *p* = 0.729). A negative trend between appendicular skeletal muscle mass (ASMM) and *PTGER4* seems to confirm the impairment of muscle regeneration associated with sarcopenia. The expression of the studied genes revealed a trend in associations with the clinical picture of muscular dystrophy and weakening patients. Perhaps *PTGS2* and *PTGES2* is in opposition to the role of the *PTGER4* receptor in muscle physiology. Nevertheless, further, including functional studies is needed.

## 1. Introduction

Frailty is defined as a state of decreased physiological reserve and compromised capacity to maintain homeostasis. It is a result of multiple accumulated deficits associated with the aging process and with several inter-related physiological systems [1]. One of the factors contributing to the development of frailty is the loss of skeletal muscle mass and strength, defined as sarcopenia [2]. 

Both described conditions are determined by genetic and environmental factors in combination with epigenetic mechanisms, which regulate the differential expression of genes in cells and could be especially important in aging [1]. Until now, there is no known effective treatment or intervention for frailty [3]. Moreover, failure to diagnose frailty exposes patients to interventions from which they might not benefit and could be harmed [1].

Prostaglandins are well known for their physiological role in the inflammatory response [4]. Depending on the interaction with different prostaglandin E receptors (EP1, EP2, EP3, and EP4) located on the cell membrane or organelle membrane, the response and biological effect on PGE2 vary [5]. PGE2 regulates different stages of the immune response during several pathologies [6]. One of the important pathways for muscle development or regeneration is activated by prostanoid receptor EP4 (encoded by *PTGER4* gene)–prostaglandin E2 (PGE2) signaling [7]. EP4 belongs to the G-protein coupled receptor family and activates adenylyl cyclase, increasing intracellular cAMP. Moreover, prostaglandins and the cyclooxygenase (COX) pathway are also crucial in the regulation of aging and exercise adaptation in the skeletal muscle [8]. PGE2 inhibits collagen production and induces matrix metallopeptidase 1 (MMP1) [9,10] associated with aging. Minthas et al. revealed a significant association between increased synthesis of PGE2 in human monocyte-derived macrophages (MDMs) with age [11]. On the other hand, a recent study noted that aged tissues, including skeletal muscle (myofibers and interstitial cells within muscle) are characterized by high levels of 15-hydroxyprostaglandin dehydrogenase (15-PGDH), which takes part in the degradation of PGE2. Therefore, according to the authors, reduction in PGE2 signaling is a major contributor to muscle atrophy associated also with sarcopenia [12]. 

Additionally, sarcopenia and frailty syndrome are associated with tissue fibrosis, particularly in the liver and lungs [13,14]. Evidence from several human diseases and data from animal models revealed that one of the pathological fibrosis pathways leads to Wnt/β-catenin signaling [15]. β-catenin is encoded by the *CTNNB1B* gene, so-called protein moonlight, which means that its cellular functions may be radically different, from the adhesion to signaling role [16]. Nevertheless, the majority of prior studies, including its relationship with prostaglandin signaling, have focused on the oncological role of β-catenin [17,18,19,20].

Prostaglandins are synthesized from arachidonic acid, and their production depends on prostaglandin-endoperoxide synthase (PTGS, COX) activity, also known as cyclooxygenase [4] and prostaglandin E synthases (PTGES) [8]. PTGS/COX occurs in two isoforms, COX-1 and -2, where COX-2 seems more significant in prostanoid formation in inflammation and proliferative diseases [4]. In the case of PTGES, there are three forms known (microsomal—PTGES1, cytosolic—cPTGES, and membrane-associated PTGES2). The physiological role of cPTGES is not well known. PTGES1 appears to be mostly involved in inflammation [3,4,5]. On the contrary, *PTGES2* is not induced by inflammatory stimuli and is constitutively expressed in various tissues that show relatively a low expression of *PTGES1*. Therefore, it is assumed that PTGES2 is important in total PGE2 production/synthesis [8]. Muscle samples of elderly people with sarcopenia also showed alterations in ferroptosis-related factors, including prostaglandin-endoperoxide synthase 2 (PTGS2/COX2) [21]. It is suggested that ferroptosis may play a role in the pathogenesis of sarcopenia [22]. Upregulated *PTGS2* expression also has an impact on the aging process [23]. The expression of *PTGES2* similarly seems to increase with age [24]. 

In summary, prostaglandin signaling pathways are closely related to aging, muscle development and regeneration, inflammation, and other processes, such as tissue fibrosis or ferroptosis, which are associated with frailty and sarcopenia. Therefore, we assume that prostaglandins play a crucial role in the frailty syndrome and sarcopenia. The present study aimed to examine the effects of aging (healthy subjects: 25–30 and >50 years old) on the profile of *PTGS2* (*COX2*), *PTGES2*, *PTGER4*, and *CTNNB1* mRNA levels in whole blood. Furthermore, this research also aims to compare the expression profile of the above-mentioned genes and find their association with clinical parameters and selected cytokine concentrations in patients with frailty syndrome and sarcopenia. 

## 2. Results

### 2.1. Clinical Characteristics of Study Groups

All groups of analyzed patients matched the female/male ratio (*p* = 0.45). Clinical parameters were compared between three groups of patients (Table 1). If significant differences were observed, pairwise post hoc analysis was performed (Appendix A).

In the present study, patients with frailty syndrome were significantly older than geriatric control (*p* = 0.006). We also observed that the control group was characterized by higher concentrations of iron (*p* < 0.001), hemoglobin (*p* = 0.02), total cholesterol (*p* = 0.02), and LDL (*p* = 0.004) compared with frailty syndrome patients. 

The significant difference between TUG time was noted when comparing geriatric patients with sarcopenia (*p* = 0.002) (Figure 1). FI-CGA and TUG time were the lowest in geriatric patients. FI-CGA was significantly different in all three analyzed groups. The highest FI-CGA was observed in frailty patients (mean 0.36 ± 0.09). Patients with sarcopenia were classified as prefrail (mean FI-CGA value of 0.21 ± 0.11) and geriatric patients as fit (mean of 0.21 ± 0.11). 

Patients with sarcopenia were characterized by significantly lower BMI levels than geriatric control (*p* = 0.003). In the group of patients with sarcopenia, there was no patient with obesity, and half of the patients had normal weight, whereas, in the group with frailty syndrome or geriatrics, 38% and 48% were obese, respectively (Appendix A). 

According to the criteria, the ASMM parameter was significantly lower in sarcopenia patients compared with both geriatric control (*p* < 0.001) and frailty syndrome patients (*p* < 0.001, Figure 1).

Despite the fact that patients with sarcopenia had the lowest level of triglycerides, the difference was not significant. Patients of all analyzed groups were characterized by similar levels of CRP, albumin, creatinine, and B12. Fifty percent of patients with sarcopenia suffered from osteopenia, whereas only 11% and 8% of patients with frailty and geriatric control, respectively, revealed this condition. Differences in osteopenia prevalence were statistically significant. Further analysis showed that sarcopenia patients had 11.5 ((95% CI) = (2.08, 93.95), *p* = 0.009) times higher odds of osteopenia than geriatric control and 8 times higher odds ((95% CI) = (1.79, 40.78), *p* = 0.008) than frailty syndrome patients. However, both 25% of sarcopenia and frailty syndrome patients suffered from osteoporosis, but in comparison with the geriatric control group (16% of patients with osteoporosis), the ratio was not statistically significant (Figure 2). 

### 2.2. Significant Impact of Age on the Expression of the PTGER4 Gene

Healthy subjects, regardless of age (25–30 and >50 years old), were characterized by the lowest *PTGES2* expression (Figure 3B). The mRNA level of *PTGES2* was significantly the lowest in healthy subjects at the age of 25–30 years in comparison with geriatric control, patients with frailty syndrome and sarcopenia, but not healthy subjects at age above 50 years. Older healthy subjects (above 50 years old) were characterized by a significantly lower *PTGES2* mRNA level than geriatric control (*p* = 0.03). Patients with frailty syndrome revealed a lower expression of *PTGES2* than geriatric control and patients with sarcopenia; however, the difference was not statistically significant. 

The expression between both healthy groups was not statistically significant. Geriatric control and patients with frailty syndrome and sarcopenia were characterized by a similar *PTGS2* mRNA level. The analyzed groups (geriatric control and patients with frailty syndrome and sarcopenia) all had significantly different mRNA levels of *PTGS2* compared with healthy subjects at age 25–30 and above 50 years old (Figure 3A).

Healthy subjects at age 25–30 and above 50 years old were characterized by a significantly lower mRNA level of *PTGER4* in comparison with all analyzed groups of patients (geriatric control, patients with frailty syndrome and sarcopenia) (Figure 3C). Moreover, healthy subjects aged 25–30 had a significantly lower mRNA level of *PTGER4* than healthy subjects aged above 50 (*p* = 0.01). The differences in expression of *PTGER4* between geriatric control and patients with frailty syndrome and sarcopenia were not statistically significant. However, the highest median of *PTGER4* mRNA level was observed in patients with sarcopenia (Figure 3C). 

The difference in *CTNNB1* expression between both healthy groups, regardless of age, was not statistically significant (Figure 3D). In the meanwhile, compared with the healthy subjects aged 25–30 and over 50, all analyzed groups revealed a higher mRNA level of *CTNNB1*. The highest median of the *CTNNB1* mRNA level was observed in patients with sarcopenia, but it was not statistically significant compared with the geriatric control and patients with frailty syndrome. 

### 2.3. Serum Cytokines Level

The serum IFN-y, IL-1α level in all analyzed groups was below the detection level (approximately 90–100% of patients included in the analysis). IL-10 was below the detection level in about 70% of the analyzed groups (Appendix A). 

Healthy subjects at age 20–30 and above 50 years had a significantly lower level of TNF-α than all groups of patients (Figure 4C). In almost 50% of healthy subjects at age 20–30 and 26% of healthy subjects above 50 years old, the serum TNF-α concentration was below the detection level.

In almost 70% of healthy subjects at age 20–30 and 90% of healthy subjects above 50 years old, the IL-1β concentration was below the detection level. The serum IL-10 concentration was below detection in 90–97% of both groups of healthy subjects. The IL-1β and IL-10 levels were significantly higher in frailty syndrome patients compared with healthy controls.

### 2.4. Negative Trend between ASSM and the Expression of PTGER4

We did not find significant correlations between the analyzed gene expression level and selected clinical parameters, such as CRP, vitamin D3 and B12 levels, albumin concentration. 

Nevertheless, in patients with sarcopenia, we noticed a significant negative correlation between the mRNA level of *PTGES2* and *PTGS2* and serum creatine kinase (CK) concertation (Figure 5A,B). On the other hand, positive correlations between these genes and TUG were observed in sarcopenia patients (Figure 5C,D). In patients with frailty syndrome, an average significant correlation between the expression of *CTNNB1* and the FI-CGA parameter was observed (r = 0.395, *p* = 0.017). In the case of *PTGER4*, a negative trend with the ASMM parameter was noted. However, the correlation between these factors was not significant (r = −0.224, *p* = 0.484). 

Interestingly, in geriatric patients, a significant negative correlation was found between iron concentration and *PTGS2* mRNA level (r = −0.47, *p* = 0.017); a similar but statistically not significant trend was also observed in the patients with sarcopenia. 

## 3. Discussion

The self-reliance and vitality of elderly patients are one of the main challenges of modern medicine. Sarcopenia and frailty syndrome, as two of the geriatric complexities, deprive people of this opportunity and affect mortality. Currently, there are no blood tests to diagnose sarcopenia, which would be particularly important in the early stages of the disease.

Considering gene expression as a potential biomarker, two factors should be taken into account, i.e., age and pathogenesis. These factors may affect independently, and sometimes they can complement each other, leading to, e.g., a synergistic effect. A recent analysis revealed that changes in gene expression and its dynamics in blood cells are age related compared with other tissues [25]. 

Sarcopenia and frailty are two geriatric syndromes, sharing partially a phenotype. Loss of muscle mass and function associated with aging itself is defined as primary sarcopenia, which usually precedes frailty. Frailty, according to the phenotypic model, includes exhaustion, weakness, and slowness, whereas sarcopenia combines the mass and function of the muscle. Frailty is age related, while sarcopenia is also associated with illness, starvation, and disuse. In general, there is an overlap in the criteria for these two conditions, but frailty requires weight loss, while sarcopenia requires muscle loss [26]. Only a few studies have shown the potential role of the prostaglandin pathway and its impact on these conditions [26,27,28,29], but according to our knowledge, this is the first study that focuses on gene expression in whole blood. 

Interest in biomarkers as rapid and quantitative measures in all areas of biomedical research, as well in in the field of aging and age-related diseases, has increased. In the present study, all the analyzed genes, excluding *PTGER4*, were characterized by a lower mRNA level in healthy subjects (both in the age group 25–30 years and in the group above 50 years). Geriatric patients without sarcopenia or frailty syndrome had a significantly upregulated gene expression compared with both groups of healthy subjects. The present study revealed only a **significant impact of age** on the expression of the *PTGER4* gene. Healthy subjects at age 25–30 had significantly downregulated this gene compared with healthy subjects at age above 50. Although we did not find a similar study to ours in the literature, the work of Acevedo et al. may suggest similar conclusions. The authors described age-related methylation in *PTGER2* and *PTGER4*. The methylation in leukocytes and, thus, the possible activity of these genes decreased with age [30]. Despite the fact that the common denominator is age, the current study showed an increase with age, but our analysis was performed in whole blood and not in individual subpopulations of cells. Unfortunately, the present study does not examine serum PGE2 or PGDH2 concentration, which could have shown a more complete picture of this pathway. 

Patients with sarcopenia were characterized by the highest expression of *PTGER4* and *CTNNB1* in whole blood samples. However, the impact of the analyzed conditions/diseases on the mRNA level of these genes was not significant. More interestingly, a negative trend between the appendicular skeletal muscle mass parameter (ASSM) and the expression of *PTGER4* was noted. Ho et al. revealed that the major effect of PGE2 during muscle regeneration is on satellite cells (muscle-specific stem cells) and that this effect is direct and mediated by the EP4 receptor encoded by the *PTGER4* gene [31]. The negative trend of ASMM and *PTGER4* seems to confirm the impairment of muscle regeneration associated with sarcopenia. Research on animal models shows poor regeneration of muscles with age [32] and the reduction of the satellite cells’ ability to self-renew [33]. According to a study conducted by Brack et al., differentiation of muscle cells and fibrosis are also associated with Wnt signaling [34], in which CTNNB1 may be implemented. Florian et al. described the reduced protein levels of β-catenin (CTNNB1) upon aging and switching from the canonical Wnt signaling to noncanonical Wnt signaling in aging hematopoietic stem cells [35], which may, in general, result in the prevention of the self-renew ability. Additionally, in patients with sarcopenia, a negative correlation was also observed between creatine phosphokinase and the expressions of *PTGES2* (*COX2*) and *PTGS2*. An increase in CK in the bloodstream indicates muscular damage [36]. The expression of these genes was also positively correlated with the functional mobility test TUG in patients with sarcopenia. Despite the fact that the generalizability of the expression level of single genes with a complex functional test seems to be inadequate, the clinical picture presented thus far supports the idea that PTGES and PTGS2 may play a role in the pathology of sarcopenia. If the roles of PTGES2 and PTGS2 in relation to the receptor EP4 (PTGER4) are opposite in muscle physiology (degeneration, ferroptosis versus regeneration), then the question remains, why does receptor activity increase with age? Is this a homeostatic mechanism of physiology? Is it just a result of the medications taken?

The present study also revealed a negative association between the iron level in blood and the expression of *PTGS2* in geriatric patients. A similar, but statistically not significant, tendency was also noted in patients with sarcopenia. Interestingly, PTGS2 is considered to be a genetic hallmark of ferroptosis—a novel form of iron-dependent programmed cell death forms [37]. One of the new approaches to treating iron-related disorders is iron chelators, which effectively prevent ferroptosis [38]. Since the studies conducted by Yang et al. and Sun et al, an upregulated mRNA *PTGS2* level is reported to be a pharmacodynamic marker of ferroptotic tissues. However, studies do not indicate that it is a contributing factor to ferroptosis [39,40]. As Chen et al. rightly pointed out, the great challenge for PTGS2 as a biomarker of ferroptosis is that the upregulation of PTGS2 is observed in various inflammatory conditions, at least some of which are nonferroptotic [41]. Recently, Xu et al. concluded that after cerebral ischemia reperfusion injury, the inhibition of ferroptosis inactivates the COX-2/PGE2 pathway [42]. Another aspect of this correlation is the source of serum iron in patients from different groups (patients with or without sarcopenia). In general, serum iron levels should not be used to diagnose iron deficiency but serum ferritin level [43]. Iron absorption is upregulated by iron deficiency and increased erythropoiesis and downregulated in inflammation [44]. It is well known that iron deficiency in the skeletal muscle is associated with reduced a concentration of myoglobin, which has an impact on tissue physiology. In terms of whole blood analysis and immune cells, iron is a key component of the enzymes that generate superoxide and nitric oxide, which are critical for the proper enzymatic functioning of cells [45]. Furthermore, iron is involved in the regulation of cytokine production in T cells [46]. In the context of sarcopenia and frailty syndrome, low iron blood serum concentrations are thought to be associated with poor physical performance [45]. 

Moreover, patients with frailty syndrome and sarcopenia were characterized by a higher serum TNF-α concentration than geriatric patients; however, the difference was not significant. We found a correlation between *PTGES2* expression and serum TNF-α concentration in geriatric patients. Despite the fact that several studies have shown that TNF-α induces PGE2 production via PTGS2 (COX2) in different tissues [47,48,49], the particular role of PTGES2 remains unclear. So far, studies determining the effect of TNF-α on *PTGES2* expression in in vitro models are inconclusive [50,51]. 

In the majority of the subjects from the healthy groups, the tested serum cytokines were not detected. The present study also did not reveal significant differences in serum cytokine level between geriatric patients and patients with frailty syndrome or sarcopenia. 

A major source of limitation of the present study is the sample size, particularly for patients with sarcopenia. Another limitation of this research is the very heterogenous group of geriatric patients who suffered from various diseases and were treated with multiple medications, which may have an impact on gene expression. Moreover, different factors associated with lifestyle, such as smoking and diet, were also not taken into account during the analysis. Therefore, further studies on a larger sample size and protein level analysis will need to be undertaken.

## 4. Materials and Methods

### 4.1. Patients

A total of 166 participant samples were included in the study. Participants in the groups of sarcopenic patients (N = 12), frailty syndrome patients (N = 36), and geriatric control (patients without sarcopenia or frailty syndrome, N = 25) were recruited in the Geriatrics Clinic and Polyclinic, National Institute of Geriatrics, Rheumatology, and Rehabilitation in Warsaw, Poland. EDTA–whole blood samples were collected and transported immediately to the laboratory and stored at −80 °C until further analysis. Blood from participants in groups of healthy subjects at age 25–30 (N = 51, first healthy control) and above 50 years old (N = 42, second healthy control) was obtained from the blood donation facility. 

Patients with sarcopenia enrolled in the study met the classification criteria of the European Working Group on Sarcopenia in Older People 2 (EWGSOP2) for this disease, which include low muscle strength and quantity. Low muscle strength was evaluated by the hand dynamometer measurement of grip strength (cut-off points for women < 16 kg, for men < 27 kg). Low muscle quantity was assessed by the measurement of appendicular skeletal muscle mass (ASMM) by dual-energy X-ray and calculated as ASMM/height^2^ (cut-off points for women < 5.5 kg/m^2^, for men < 7 kg/m^2^) [52].

Patients with frailty syndrome included in the study met the classification criteria of the frailty index based on a standard comprehensive geriatric assessment (FI-CGA—frailty index based on a comprehensive geriatric assessment). This test is based on the evaluation of 10 domains: cognitive status, mood and motivation, communication, mobility, balance, bowel function, bladder function, instrumental activity of daily living and activity of daily living (IADLs and ADLs), nutrition, and several noncommunicable chronic diseases [53].

Patients diagnosed with both sarcopenia and frailty syndrome were not included in the study. Participants from the geriatric control group were patients who were 60 years old and above without sarcopenia or frailty syndrome. The functional mobility of the patients was described using the timed up and go scale (TUG). This test evaluates the ability to rise up from the chair, walk, return to the chair, and sit down again. Additionally, the task checks the walking speed, strength, balance, and substantial cognitive involvement. All patients were also screened for osteoporosis, osteopenia, and degenerative disc disease. Additionally, the New York Heart Association (NYHA) functional classification system was used to assess evidence of cardiovascular disease (class I—no objective evidence of cardiovascular disease; class II—objective evidence of minimal cardiovascular disease; class III—objective evidence of moderately severe cardiovascular disease; class IV—objective evidence of severe cardiovascular disease).

People with exacerbations of inflammatory diseases, active infections, diarrhea, and thrombotic diseases and those treated with prostaglandin analogues or systemically administered anti-inflammatory drugs were excluded from the study.

All participants provided their written informed consent before enrollment. This study was performed in accordance with the Declaration of Helsinki and was approved by the Research Ethics Committee of the National Institute of Geriatrics, Rheumatology, and Rehabilitation (approval number KBT-4/2/2018). 

### 4.2. RNA Extraction

Total RNA was extracted from 500 uL of whole blood using a Micro RNA Concentrator (A&A Biotechnology, Gdańsk, Poland) with TRIzol Reagent (Invitrogen, Foster City, CA, USA). The quantity and quality of isolated RNA was evaluated by a Quawell Q5000 spectrophotometer. cDNA was prepared using High Capacity cDNA Reverse Transcription with an RNase Inhibitor Kit (Life Technologies, Foster City, CA, USA), according to the manufacturer’s instruction.

### 4.3. Gene Expression

Gene expression analysis was conducted using TaqMan Gene Expression Assays designed by the manufacturer (Thermo Fisher, Foster City, CA, USA): *PTGES2* (Hs00228159_m1; prostaglandin E synthase 2), *PTGER4* (Hs00168761_m1; prostaglandin E receptor 4), *PTGS2* (*COX2*) (Hs00153133_m1; prostaglandin-endoperoxide synthase 2), *CTNNB1* (Hs00355045_m1; β-catenin), and TaqMan Gene Expression Master Mix (Thermo Fisher, Foster City, USA). The manufacturer guaranteed the PCR efficiency of all assays at 100 ± 2%. Each sample was analyzed in two technical replications, and the mean Ct value was taken for further analysis. A Ct value higher than 35 was taken as below quantification. The most relevant housekeeping gene was selected, and the relative expression was calculated by the ΔCt method (normalized to *GAPDH* (Hs02786624_g1) as a reference gene using QuantStudio 5 Real-Time PCR System (Applied Biosystems, Foster City, CA, USA). 

### 4.4. Serum Cytokine Level

TNF-α, IFN-γ, IL-1a, IL-1b, IL-6, and IL-10 concentrations in the serum of patients with frailty syndrome, sarcopenia, and geriatric control and healthy subjects at age 25–30 (first healthy control) and above 50 years old (second healthy control) were measured by multiplex enzyme-linked immunosorbent assay (ELISA), Q-Plex Human Cytokine Panel 1 (6-Plex), according to the manufacturer’s instructions (Quansys Biosciences, Logan, UT, USA). For analysis, we used 25 uL of serum diluted 1:2 with a sample diluent. The ELISA standard level of detection was 10.42 pg/mL for IL-1b, 35.12 pg/mL for IL-6, 20.86 pg/mL for IFN-γ, 21.68 pg/mL for IL-1a, 17.28 pg/mL for IL-10 (pg/mL), and 3.98 pg/mL for TNF-α. For analysis, patients with values below the level of detection were analyzed in two variants: having a cytokine level equal to 0 or equal to the detection threshold. Only consistent results from the two analyses were reported.

The Q-Plex Human Cytokine Panel 1 (6-Plex) assay is an ELISA-based test premised on nano spot technology. In each well of a microtiter plate, immobilized capture antibodies are attached to the surface of several nano spots. Capture antibodies sealing each nano spot are specific for only one analyte. This technology enables the measurement of multiple analytes in one well of a 96-well plate, provides high sensitivity of the assay, and requires a minimal sample volume. 

The basic steps of Q-Plex are similar to the traditional ELISA test. It involves samples or calibrator incubation, washing, incubation with biotinylated antibodies, washing, incubation with an enzyme (streptavidin–horseradish peroxidase (SHRP)), washing, and incubation with a substrate. The detection of the chemiluminescence signal of the product was performed using Q-View Imager LS and Q-View Software (Quansys Biosciences, Logan, UT, USA). Analysis of cytokine levels between groups assumed that patients with levels below the detection level had a level equal to the minimum threshold value.

### 4.5. Statistical Analysis

Missing data are summarized in a table in the Appendix A. Data were imputed assuming data missing at random MAR. Missing data were imputed using random forest.

The normality of the distribution of the analyzed parameters was checked by the Shapiro–Wilk test. Demographic and clinical parameters between analyzed groups were compared using ANOVA, Welch’s ANOVA, and the Kruskal–Wallis rank sum test for continuous variables. Discrete variables were analyzed using Pearson’s chi-squared test or Fisher’s exact test for count data. A significant post hoc analysis was performed to identify pairwise differences. The *p*-value for multiple comparisons was corrected using the Bonferroni–Holm method. The statistical significance of differences in gene expressions and serum cytokine levels between groups was analyzed using the Kruskal–Wallis test and Dunn’s multiple comparison test. The correlation between variables was evaluated using the Spearman test. RStudio Version 1.4.1717 © 2009–2021 RStudio, PBC, was used to conduct analysis and present graphs. The R packages used for data analysis are listed in the References section [54,55,56,57,58,59,60].

## 5. Conclusions

The level of the studied genes revealed an association trend with the clinical picture of muscular dystrophy, weakening patients. Further studies would have to test whether *PTGS2* and *PTGES2* in muscle physiology are in opposition to the PTGER4 receptor. 

## Figures and Tables

**Figure 1 ijms-24-12885-f001:**
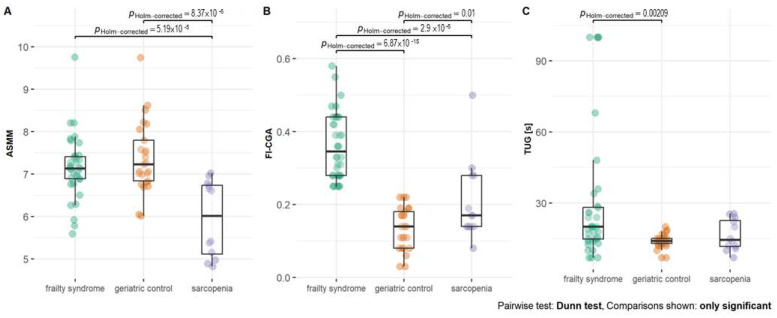
(**A**) Appendicular skeletal muscle mass (ASMM), (**B**) frailty index based on a comprehensive geriatric assessment (FI-CGA), and (**C**) timed up and go scale (TUG) parameters—functional mobility test—in patients with frailty syndrome, sarcopenia, and geriatric patients without frailty and sarcopenia as a control. Significance at *p-value* < 0.05.

**Figure 2 ijms-24-12885-f002:**
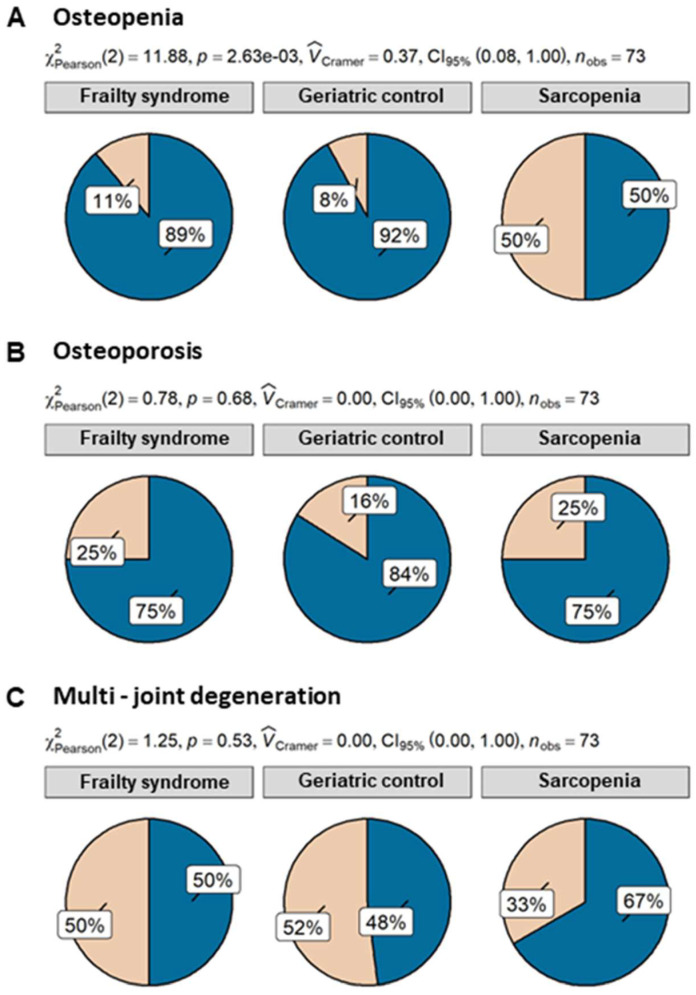
Percentage of osteopenia (**A**), osteoporosis (**B**), and multijoint degeneration (**C**) in patients with sarcopenia (n = 12), frailty syndrome (n = 36), and geriatric control group (n = 25). Legend: beige—present; blue—not present.

**Figure 3 ijms-24-12885-f003:**
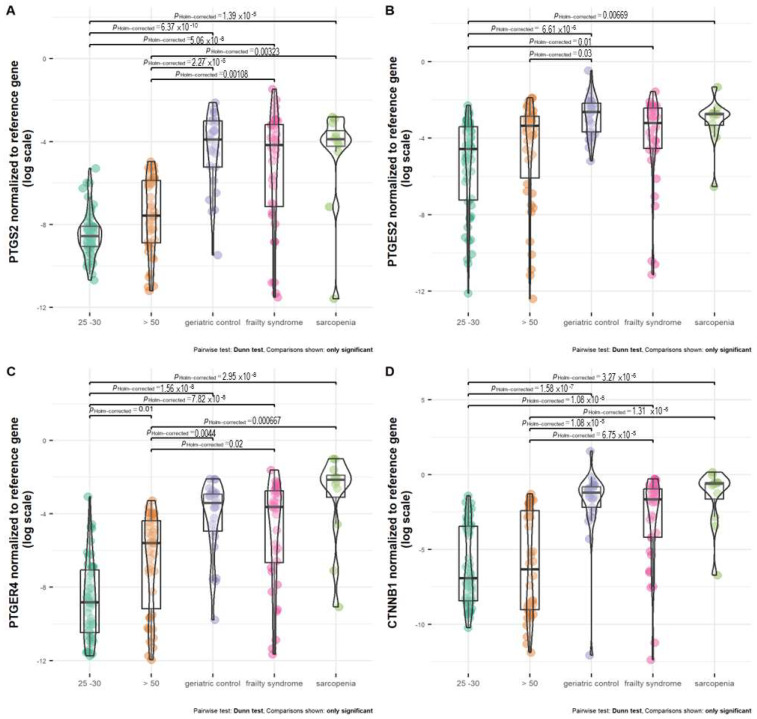
Expression of (**A**) *PTGS2*, (**B**) *PTGES2*, (**C**) *PTGER4*, and (**D**) *CTNNB1* normalized to reference gene in healthy subjects at age above 50 (>50), 25 –30 years old, and patients with frailty syndrome, sarcopenia, and geriatric control.

**Figure 4 ijms-24-12885-f004:**
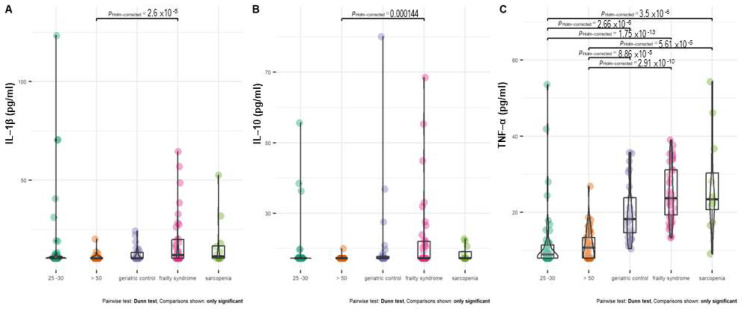
Distribution of IL–1 β (**A**), IL–10 (**B**), and TNF–α (**C**) level between treatment groups. In the case of patients with a cytokine concentration below the limit of detection, the detection threshold value was imputed.

**Figure 5 ijms-24-12885-f005:**
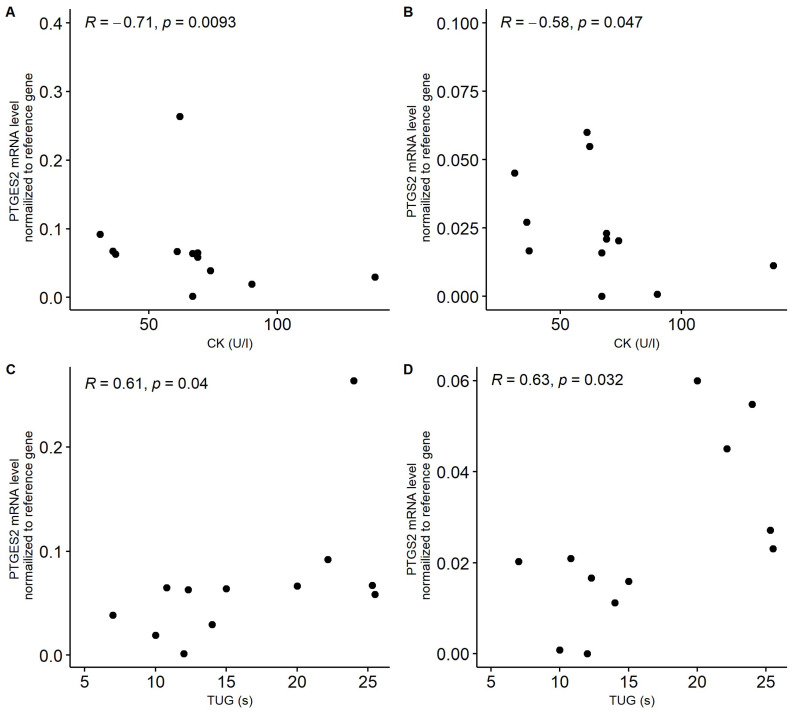
Correlation between serum creatinine kinase (CK, U/L) concentration and expression of PTGES2 (**A**) and *PTGS2* (**B**) in patients with sarcopenia. Correlation between functional mobility test TUG parameter (timed up and go scale) and expression of *PTGES2* (**C**) and *PTGS2* (**D**) in patients with sarcopenia. Gene expression was normalized to reference gene.

**Table 1 ijms-24-12885-t001:** Demographic and clinical characteristics of geriatric control and patients with frailty syndrome and sarcopenia.

	Frailty Syndrome (N = 36)	Geriatric Control (N = 25)	Sarcopenia (N = 12)	*p*-Value
Age (years, mean ± SD)	80.50 ± 8.72	72.56 ± 7.86	77.33 ± 13.75	0.004 ^a^
Gender: Women n (%) Men n (%)	25 (69.44%) 11 (30.56%)	17 (68.00%) 8 (32.00%)	6 (50.00%) 6 (50.00%)	0.45 ^c^
	mean ± sd	mean ± sd	mean ± sd	*p*-value ^a^
ASMM (kg)	7.19 ± 0.74	7.47 ± 0.80	5.95 ± 0.90	<0.0001
FI-CGA	0.36 ± 0.09	0.14 ± 0.06	0.21 ± 0.11	0.004
BMI (kg/m^2^)	27.61 ± 6.37	29.86 ± 5.28	23.18 ± 2.97	0.005
Cholesterol (mg/dL)	173.85 ± 41.31	204.24 ± 42.99	194.50 ± 42.75	0.022
Triglycerides (mmol/L)	138.74 ± 66.59	147.04 ± 60.18	105.58 ± 42.91	0.154
LDL (mg/dL)	91.06 ± 30.08	120.50 ± 37.65	100.65 ± 37.64	0.006
Albumin (g/dL)	3.98 ± 0.49	4.10 ± 0.32	4.17 ± 0.38	0.289
Hemoglobin (g/dL)	12.11 ± 1.73	13.29 ± 1.43	12.50 ± 1.60	0.023
LDH (U/L)	252.89 ± 103.51	229.81 ± 41.67	260.52 ± 61.35	0.443
Creatinine (µmol/L)	0.88 ± 0.24	0.89 ± 0.27	0.97 ± 0.29	0.605
Vit. B12 (pg/mL)	397.72 ± 187.11	426.20 ± 243.40	455.65 ± 245.43	0.704
Serum Fe level (mcg/dL)	61.16 ± 22.84	84.11 ± 22.97	79.96 ± 10.50	<0.0001
	median (IQR)	median (IQR)	median (IQR)	*p*-value ^b^
TUG (s)	20.00 (14.75, 26.50)	14.00 (13.00, 15.00)	14.50 (11.70, 21.97)	<0.0001
ESR (mm/h)	22.00 (12.50, 35.25)	10.00 (8.00, 22.00)	11.00 (8.00, 15.50)	0.048
CRP (mg/dL)	7.00 (5.00, 25.50)	5.00 (5.00, 12.00)	5.00 (5.00, 6.00)	0.062
NT-proBNP (pg/mL)	264.30 (163.67, 510.20)	258.60 (164.60, 559.60)	354.80 (277.88, 666.57)	0.308
CK (U/L)	71.00 (45.25, 86.50)	72.00 (56.00, 106.00)	67.00 (55.00, 70.79)	0.382
	n (%)	n (%)	n (%)	*p*-value ^c^
Osteopenia				0.009
Present	4 (11.11%)	2 (8.00%)	6 (50.00%)	
Not present	32 (88.89%)	23 (92.00%)	6 (50.00%)	
NYHA class				0.153
0	27 (75.00%)	23 (92.00%)	7 (58.33%)	
I	1 (2.78%)	0 (0.00%)	0 (0.00%)	
II	6 (16.67%)	1 (4.00%)	3 (25.00%)	
II-III	1 (2.78%)	1 (4.00%)	2 (16.67%)	
III	1 (2.78%)	0 (0.00%)	0 (0.00%)	
Decrease				0.213
Yes	7 (19.44%)	2 (8.00%)	0 (0.00%)	
No	29 (80.56%)	23 (92.00%)	12 (100.00%)	
Polyarthritis				0.574
Present	18 (50.00%)	13 (52.00%)	4 (33.33%)	
Not present	18 (50.00%)	12 (48.00%)	8 (66.67%)	
Osteoporosis				0.691
Present	9 (25.00%)	4 (16.00%)	3 (25.00%)	
Not present	27 (75.00%)	21 (84.00%)	9 (75.00%)	
Degenerative disc disease				0.945
Present	13 (36.11%)	8 (32.00%)	4 (33.33%)	
Not present	23 (63.89%)	17 (68.00%)	8 (66.67%)	

ASMM—appendicular skeletal muscle mass; FI-CGA—frailty index based on a comprehensive geriatric assessment; BMI—body mass index; LDL—low-density lipoprotein; LDH—lactate dehydrogenase; Fe—iron; TUG—timed up and go scale (mobility scale); ESR—erythrocyte sedimentation rate; CRP—C-reactive protein; NT-proBNP—N-terminal (NT)-pro hormone BNP; CK—creatinine phosphokinase; NYHA class—New York Heart Association class; ^a^ one-way analysis of means (not assuming equal variances) or analysis of variance; ^b^ Kruskal–Wallis rank sum test; ^c^ Pearson’s chi-squared test or Fisher’s exact test for count data.

## Data Availability

Data available on request.

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
