# Peer review of "Expression of Prostaglandin Genes and β-Catenin in Whole Blood as Potential Markers of Muscle Degeneration"

_ijms, 2023, doi:10.3390/ijms241612885_

Round 1

Reviewer 1 Report

Overall, this is a nicely done and well written publication. The study design is appropriate and apparently, the analyses were carefully performed. This manuscript shows rich and valuable content, which is within the journal’s scope. However, before publication some points need to be clarified.

My comments:

Line 10, 51, 67 and the rest of the manuscript – genes names should be written in italics.

Line 87 - please present your hypothesis correctly.

Line 225 – goals of the study were already presented in the Introduction chapter.

Line 333 – please characterize more how the blood was stored in blood donation facility. What was the kind of blood donation (whole blood, power red donation or other)?

Line 339 – DXA acronym is used only one time. I see no sense to abbreviate it.

Line 372 – how the authors designed and synthesized primers?

Line 372 – table with primers used is missing.

Line 381 – I think authors should use two reference genes.

Line 426 – Please avoid any speculation in conclusion. Also, some future perspective should be added.

Author Response

Thank you for the detailed review and valuable comments.

  • Line 10, 51, 67 and the rest of the manuscript – genes names should be written in italics.

As the Reviewer suggested,  it has been corrected. In the context of general mechanisms or protein level we left standard writing.

  • Line 87 - please present your hypothesis correctly.

As the Reviewer suggested, we clarify our hypothesis: “(…) we assume that prostaglandins play crucial role in the frailty syndrome and sarcopenia”.

  • Line 225 – goals of the study were already presented in the Introduction chapter.

Goals of the study have been removed from the discussion chapter.

  • Line 333 – please characterize more how the blood was stored in blood donation facility. What was the kind of blood donation (whole blood, power red donation or other)?

It was EDTA whole blood stored at -80. Both, patients’ and healthy donors’ blood samples were taken, processed and stored at the same conditions. The information has been added into material and methods section.

  • Line 339 – DXA acronym is used only one time. I see no sense to abbreviate it.

As the Reviewer suggested,  DXA acronym has been removed.

  • Line 372 – how the authors designed and synthesized primers? Line 372 – table with primers used is missing.

TaqMan Gene Expression Assays were designed by manufacturer (Thermofisher). We choose assays with the best coverage, citation and spans exons, if available. Based on presented probes ID in the manuscript, every reader can choose the same probes.  

  • Line 381 – I think authors should use two reference genes.

In accordance with the recommendations, we know that at least two reference genes should be used for the study. However, because of the limited access to biological material, we decided to use a single reference gene. The reference gene chosen was the most stable and reproducible in all studied groups.

  • Line 426 – Please avoid any speculation in conclusion. Also, some future perspective should be added.

As the Reviewer suggested,  it has been corrected.

Reviewer 2 Report

Manuscript ID: ijms-2565838

Title: Expression of prostaglandin genes and β-catenin in whole blood as potential markers of muscle degeneration

Journal: International Journal of Molecular Sciences

The authors have investigated the expression of prostaglandin genes and β-catenin in whole blood as a potential marker of muscle degeneration. In this work, the authors investigated the negative trend between ASMM and PTGER4, confirming the impairment of muscle regeneration associated with sarcopenia. Expression of the studied genes reveals a trend in associations with the clinical picture of muscular dystrophy and weakening patients. Thus, PTGS2 and PTGES2 may be in opposition to the role of the PTGER4 receptor in muscle physiology. However, some details need further explanationThe article is well-written and well-organized, and I believe that it is suitable for publication in the International Journal of Molecular SciencesHowever, I recommend this manuscript can be published after a minor revision and improvement.

1. The author should follow our journal "International Journal of Molecular Sciences" formatting style.

2. Title should be "Expression of prostaglandin genes and β-catenin in whole blood as potential markers of muscle degeneration."

2. Abstract

The abbreviations used in the abstract should be consistent in an abbreviate form on their first expression throughout the manuscript; however, the full term and abbreviate have been used more than one time on their first expression. Please, the author needs to be consistent in this regard.

3. Introduction

1. Regarding the abbreviation, the same inconsistency is happening here again.

4. Results

1. The author should cite Table S1 in the main manuscript body.

2. Table S2 instead of Table 2S.

3. The table is better with 3 lines, such as 2 on the top and 1 at the bottom. Follow the rules to upgrade the manuscript to the next level.

4. Specify the Figure 1 part, such as A, B, and C, in the caption with an elaborate sub-caption for each.

5. Figure 1 numbering has been repeated two times. Please, double-check as it should be Figure 3.

6. In Figure 5, part B is missing in the caption. Provide it.

5. Please correct typos, errors, and grammar throughout the manuscript; also, the manuscript needs consistency regarding writing and formatting. The manuscript needs to be edited and proofread by a professional editor as the language needs improvement.

The manuscript needs to be edited and proofread by a professional editor as the language needs improvement.

Author Response

Thank you for the detailed review and valuable comments.

  1. The author should follow our journal "International Journal of Molecular Sciences" formatting style.
  2. Titleshould be "Expression of prostaglandin genes and β-catenin in whole blood as potential markers of muscle degeneration."
  3. Abstract: The abbreviations used in the abstract should be consistent in an abbreviate form on their first expression throughout the manuscript; however, the full term and abbreviate have been used more than one time on their first expression. Please, the author needs to be consistent in this regard.
  1. Introduction: Regarding the abbreviation, the same inconsistency is happening here again.

With regard to abovementioned comments (1-3), the manuscript has been adjusted and the abbreviation has been made consistent.

 Results

  1. The author should cite Table S1 in the main manuscript body.

Table S1 is cited in the Material and methods chapter in the Statistical analysis section: “Missing data are summarized in a table in Supplementary material (Table 1S).”

 Table S2 instead of Table 2S.

As the Reviewer noted, it has been corrected.

  1. The table is better with 3 lines, such as 2 on the top and 1 at the bottom. Follow the rules to upgrade the manuscript to the next level.

As the Reviewer noted, it has been corrected.

  1. Specify the Figure 1 part, such as A, B, and C, in the caption with an elaborate sub-caption for each.

As the Reviewer suggested, it has been specified.

  1. Figure 1 numbering has been repeated two times. Please, double-check as it should be Figure 3.

The figure numbering has been checked and corrected.

  1. In Figure 5, part B is missing in the caption. Provide it.

As the Reviewer noted, it has been corrected and bolded to make it more visible.

 5.Please correct typos, errors, and grammar throughout the manuscript; also, the manuscript needs consistency regarding writing and formatting. The manuscript needs to be edited and proofread by a professional editor as the language needs improvement.

The manuscript has been revised also for the English Language. All applied changes are highlighted.